# Ecological Grief Observed from a Distance

**Ondřej Beran** 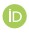

Department of Philosophy and Religious Studies, University of Pardubice, 532 10 Pardubice, Czech Republic; ondrej.beran@upce.cz

**Abstract:** The paper discusses ecological grief as a particular affective phenomenon. First, it offers an overview of several philosophical accounts of grief, acknowledging the heterogeneity and complexity of the experience that responds to particular personal points of importance, concern and one's identity; the loss triggering grief represents a blow to these. I then argue that ecological grief is equally varied and personal: responding to what the grieving person understands as a loss severe enough to present intelligibly a degradation of her life and the world, to their meaningfulness or even sustainability. More specifically, both personal and ecological grief may manifest in an eroded sense of the future as a space in which one would invest oneself with plans, projects, ideas, desires, and endeavours. On the other hand, personal grief is, in some cases, conceptualised as having embedded the inherent possibility to come to closure or "move on" (e.g., by marrying again), while with ecological grief, the intelligibility of overcoming (replacing) the loss may be, depending on its scale, severely limited. I argue that this erosion of the future need not take the shape of paralysing sadness but rather of a disruption of taking some options of projecting oneself into the future seriously or as real.

**Keywords:** ecological emotions; anticipatory grief; the erosion of the future; moving on

## 1. Introduction

"There's nothing [that] makes me more depressed than to see the place—dust lifting off the place. It's really terrible [...] I can't stand the place blowing away. Dust! I get in bed and pull the rugs over my head so I can't see it." [1] (p. 276)

"Inuit are people of the sea ice. If there is no more sea ice, how can we be people of the sea ice?" (p. 277)

These two voices, belonging to an Australian farmer and to an Inuit respondent from Nunatsiavut (Labrador, Canada), respectively (as reported by Ashlee Cunsolo and Neville Ellis [1]), express the mournful relationship of the speaker to a disturbed, damaged place that has mattered to them. In Cunsolo and Ellis's study, these voices represent expressions of what the authors call "ecological grief":

the grief felt in relation to experienced or anticipated ecological losses, including the loss of species, ecosystems and meaningful landscapes due to acute or chronic environmental change. (p. 275)

In view of the harsh reality of the events triggering ecological grief, Cunsolo and Ellis see it as a "natural response" rather than as neurotic or misplaced. Similar descriptions have been offered under slightly different names. Rupert Read [2] (p. 102f) speaks of "climate grief" or "extinction grief", a response to the shattering doubt cast on the certainty of life, to the "ongoing rip in the fabric of our shared world". For Read, grief (climate grief included) is a transformation of love, a response to the loss of someone or something we have loved, only here it is not love for a person but for nature, animals and plants, or places (cf. also [3]). Kriss Kevorkian [4] (p. 217) speaks of "environmental grief" but moves along similar lines: "the grief reaction stemming from the environmental loss of ecosystems caused by either natural or human-made events." In what follows, I will stick with the term "ecological

grief". While I would personally, for reasons of a vaguely "aesthetic" nature, probably prefer "environmental grief", "ecosystem" is the most inclusive term, encompassing and integrating both "climate" and "environment", in addition to its inhabitants. Speaking of "ecological grief" thus allows for the widest variety of cases to be covered, responding to very heterogeneous particular triggers.

Ecological grief was originally *noticed* as a phenomenon specific to environmentalists and naturalists (people professionally positioned to be close to and to care about the ecosystems, biological species, or habitats they study) [5]—but cf. also [6] as a testament that this is still a worrying mental health trend among these professionals. The roots of the phenomenon are deeper and broader, though. In the Western context, precursors of this affective experience as something more universal can be found in the classics of environmentalist writing by Aldo Leopold [7] and Rachel Carson [8] but also as far as Romantic poetry (cf. Pihkala's [9] (Section 1.1) historical overview). For Indigenous people, though, ecological grief, if unnamed, has long been a part of the lived reality, as they faced colonialist oppression and expulsion from their traditional lands (exploited for the purpose of the colonisers' agriculture or mining) [10]. The early environmentalist writings serve as early expressions of the observation that we are all embedded in our environments and respond to changes and processes within them. This is the reason why ecological grief has, finally, become increasingly recognised and studied as an endemic experience beyond the circles of scientists and environmentalists, as "global environmental change and regional ecological decline are increasingly embedded within everyday experience" [1].

In this paper, I want to elaborate on the existing research and its illuminating observations by trying to show that a mournful reaction to the loss of valued ecosystems or natural places may have a complicated, less-than-obvious structure and that it makes good sense to understand some reactions to the climate predicament as cases of ecological grief, although they look unlike "typical" cases of grieving on the surface. Compared to the works by Cunsolo or Kevorkian, my discussion is more specifically philosophical, but there is, I believe, a natural continuity. In Section 2, I will first (in Section 2.1) briefly introduce some important philosophical accounts of grief in recent literature and then (in Section 2.2) relate a few more personal accounts of grief, exploring its irreducibly personal nature (C. S. Lewis, Denise Riley, Rush Rhees). In Section 3, I will argue how the extrapolation of these accounts allows us to conceptualise ecological grief as a personal and personalised response to how the one who experiences it perceives and understands the ecological predicament. Accounts of emotion as a form of understanding will support this exploration in, I believe, an indispensable manner, though I will not argue in favour of any form of cognitivism as a comprehensive general theory of emotion. In Section 4, I will focus on one point of partial difference between ecological grief and grief in personal contexts: ecological grief, in some of its forms, displays intrinsically an eroded possibility of "moving on", stemming from the (temporally) different forms of the eroded sense of the future. I will, again, plead for making sense of such forms of ecological grief in terms of, at least partly, understanding and perception, in order to be able to do justice to this eroded possibility. The forms of ecological grief I am interested in and the observations I will make about the eroded sense of the future concern more neatly cases of global, overwhelming grief rather than local and particular; my points thus need to be taken into account with this proviso (I will try to caution the readers to the limits of my account throughout).

## 2. Accounts of Grief

### 2.1. Some Important Comprehensive Accounts of Grief

A very good discussion of the varied facets of grief is offered by Michael Cholbi in his recent book [11]. Cholbi introduces his discussion by carefully acknowledging the *heterogeneity* of the grief experience: we grieve for many different kinds of significant others and we grieve in many different ways that often do not resemble each other much. However, Cholbi strives to offer a unitary account of grief, arguing that exactly because of the heterogeneity of experienced cases, the unity would be the matter of the structural

aspects of the experience on the part of the experiencer. Thus, according to Cholbi, it makes little sense to identify grief as a single emotion or mood, because the phenomenology of grief differs strongly. Instead, it is a sequence of emotional states or episodes—which may or may not resemble the familiar Kübler-Ross's scheme—interconnected with resulting ways of behaviour. Cholbi finds various perhaps surprising but plausible unifying traits:

(i) The *egocentricity* of grief—we grieve for those "who play key roles in how we see ourselves and our lives, those in whom we have invested our hopes and in whom we thereby invested our practical identities" (p. 36);

(ii) The *relationality* of grief: "essential to grief, I propose, is that one's relationship with the deceased cannot continue in precisely the same guise it had when the deceased was still alive" (p. 55);

(iii) The *active* character of grief ("grief is something we do, rather than something that happens to us" (p. 45);

and, perhaps most importantly,

(iv) Grief being "a paradigm case of emotionally catalyzed *attention*" (p. 48; my emphasis).

Attention, in turn, is not a single mental state but a complex endeavour of all our various mental and psychological capacities, and it is the eventual subsiding of this exercise of attention that marks the ending of grief. This last point will have importance for my discussion later on.

Another synthesising account, complementary to Cholbi's in a fortunate way, has been offered at almost the same time by Matthew Ratcliffe [12], who also agrees that grief is a complicated process extended over time, encompassing other emotions as well. Compared to Cholbi's account—with which Ratcliffe otherwise shares various observations—less attention is perhaps paid to the first-person invested experience. As Ratcliffe suggests, grief not only incorporates various other particular emotions, but its temporality does not fit the way in which we experience these particular emotions: they (can) come and go abruptly, but grief does not seem to work like that (as Ratcliffe shows with reference to Wittgenstein's famous remarks about grief [13] [Part II, §§ 2–4]). Indeed, grief can be distinctly "gappy" in view of what, in particular, one *feels* at the moment, that is, one can still be grieving even when no particular mental or psychological experience thematising this grief (anger, sadness, thinking about the lost person, and so forth) takes place (see p. 20). Ratcliffe's discussion leads toward considering grief as something that makes a significant difference to the whole world (see Section 2.3, citing many testimonies of bereaved people; or Chapter 4 of Ratcliffe's book) for the grieving person who now lives in this changed world. The world, as Ratcliffe observes, is not simply "an object of passive experience but a context in which we are actively immersed", and this immersion takes, among others, the shape of practices that *used to be shared* practices (with those whom we have lost). The world, on account of the loss, continues only as conflicted,[1] alienated, unreal, lost, or, occasionally, not really making sense at all. Note, however, a certain tension between the world as a *whole* and Ratcliffe's explanation of our disrupted immersion into this whole through *particular* points/contexts of our experience that used to feature the lost other. This, too, will have a role in my own discussion later on.

It also seems proper and useful to mention Peter Goldie's influential account of grief as something that has a "narrative" structure [14]. Over the past decade, various objections have been raised against Goldie, and my aim is not to defend him. The rationale of his account is sound, though, as it was guided by his suspicion about theories of emotions as *episodic* mental states that happen to us and come and go. Grief, as Goldie argues, very much does *not* have this nature. More specifically, Goldie's claim is not that grief has perhaps the structure of a narrative by itself but that it is best accessible when understood through a narrative account. Grief is a process, one that features *particulars*—in fact, *series* of them—which are of *interest* to the grieving person as related to each other. The nature

of this relation is occasionally causal, too, but it is not only causal. We grieve on account of certain losses not because grief is caused and causally necessary (though this can be imagined, too) but because it makes sense: the loss justifies and motivates the grief in much the same way as events within a narrative motivate things that happen subsequently within the same narrative on account of these previous events. These sequences of events are *publicly narratable*, that is, even a third party can understand these links of motivation as intelligible. Goldie is also interested in further aspects of this narrative structure that I will leave aside; what is of more interest to me is the emphasis on the public narratability of a sequence of events as intelligibly motivating each other. The criterion of intelligibility allows for a great variety.

I do not present a particular notion of grief, either identical to any of the above or distinct from them. I will be taking grief as a complex response to a recognised loss of a value important or even central to the grieving person's life (which can subsequently assume the meaning of something left, as it were, in the past), which is a relatively unspecific and unambitious outline. I find many of the descriptions offered by Cholbi, Ratcliffe and Goldie helpful. Of particular interest to me will be the observations (made in various forms by these authors) that grief involves a transformation of what the whole world, the whole life, means to the grieving person, especially with respect to what existential options appear as open or relevant to her; some forms of ecological grief would testify to this. In particular, the situation of grief is such that it provides a logical rather than causal motivation to such a transformed experience; this link is also intelligible to a third person.

### 2.2. Some Case-Study-like Accounts

Both Cholbi and Ratcliffe explore various traits of the experience of grief and what it is like (how it is experienced) from a first-person perspective. This phenomenological approach naturally relies on first-person or autobiographical reports of grief as an invaluable resource. In this section, I will briefly introduce a few such first-person accounts, which are striking in their insight and detailed observation. A rich and illuminating "classic" of this "genre" is undoubtedly C. S. Lewis's *A Grief Observed* [15]. Lewis provides many observations on how his whole world was disrupted by the death of his wife (in line with Ratcliffe's observations, Lewis shows how the "world" is built around and of shared activities, interests, places (see p. 660f). Lewis pictures the disconcerting effect of mundane situations or encounters: either these felt inexplicably (indescribably) different from how they felt when his wife was still alive or they were not really different or differing from the normal or the expected at all, which came as perhaps an even greater surprise. Lewis's observations are perhaps a testament to how bereavement can make any subsequent encounter with the world unsettled, feeling "not quite right"—irrespective of whether the particular situation "behaves" differently. Phenomenologically, grief feels, to Lewis, in turn, like fear, suspense, concussion, or even laziness (the incapacity to invest any effort in anything). Religious thoughts often provide framing and, towards the end, a certain resolution to his pondering. Much of his notes, however, capture the conflicted and unresolved nature of the experience of grief. For instance, he captures vividly his conflicted feelings as to whether moving eventually over the pain of grief would count as a form of "betrayal": eventually escaping the pain of grief may feel like "killing the dead a second time" (p. 679) but, on the other hand, a temporary "lifting of the sorrow" feels like "removing a barrier" to true remembering, to being in touch (p. 675). The world is disordered, disoriented in the sense that it becomes unclear in which direction (in the world) this or that particular experience points. This will be of interest to us.

Michael Cholbi's observation that grief builds upon the personal investment of (the identity) the one who grieves can take various shapes. A very vivid case—testifying, in the second plan, to there being no intrinsic barrier between grieving after a human other and a non-human other—finds itself in Rush Rhees's private notes made after the death of his dog [16]. Rhees records feelings similar to those recorded by Lewis: his whole life has been

shaken in a way that makes any prospects for the future feel unintelligible or as a form of betrayal; the world has been disordered to the utmost:

> I cannot think that the world in which I move and do things now is real. The world that's real is where I left him. I've left the world, I guess; he hasn't. (p. 201)

Elsewhere, Rhees notes, literally, that "the death makes no sense"; any usual words of "wisdom" applied to this experience make no sense (p. 200). The grief narrative about a dog shows, on the one hand, that grief is intelligible and relatable in terms of a narrative, whether we talk about the case of a lost person or a non-human animal. On the other hand, Rhees's notes also show—and this observation is perhaps not always stressed enough by writers who try to capture grief theoretically in more generality—that the feelings of grief are phrased, by the one who grieves, not only in paradoxical or self-contradicting terms but that sometimes, one is simply hit by the plain meaninglessness, the absurdity of the experience:

> The last thing he understood from me was 'Down'. I've called him and called to him since then. But if 'Down' was the last thing he understood, then . . . I cannot get it straight nor understand it. (p. 207)

Later, I will suggest my own version of this observation that grief makes incomprehensible much that, as it were, "normally" proceeds.

The opaque character of grief emerges even more strongly from another rich and vivid first-person record of grief, Denise Riley's *Time Lived, Without Its Flow* [17], which is the record of the author's dealing with the loss of her adult son. Riley reports the curious experience of time having stopped in its normal sense: she is "inching along, but not forward, or in any decipherable direction" (p. 29), having no sense of a stream of time (life) towards the future, stuck in a strange kind of present that feels "paper thin"; instead of "seizing the day", feeling that "there is no time to seize" (p. 32). In effect, there is much to Riley's account that suggests a *resistance* to the possibility of providing a coherent, intelligible narrative, "now all your small mastery has been smashed by the fact of your child's death" (p. 52), and the usual terms labelling this experience (grief, mourning) do not serve this purpose, being "almost decorative" (A grieving person, if she even tries to put what she is experiencing into words, will probably *not* say, "I am grieving"). The strong (even after years of not wavering) connection of Riley to her deceased son takes the shape of both relating to him as alive and "participating in [his] complete loss of possibilities" (cf. [18], p. 204) as of a dead person. This may serve as a pointer to the fact that while grief may be talked about as something, the story of which can be understood and followed from outside, by tracing a certain narrative logic, it often (if not always) resists being *experienced* as something logical (something that has a logic) in the first person.

*2.3. A Few Disclaimers*

With the exception of Rush Rhees' diary, all accounts and cases of grief reported above concern the loss of a close *human* being. I am not taking this context as central or foundational for the capacity to experience grief in non-human contexts, though; I simply helped myself with these accounts as well-developed and illuminating. One obvious way of stressing the continuity between cases of interpersonal grief and ecological grief would be to rely on accounts of grief after non-human animals, as it is obvious that the sense of a significant loss applies to animals—not just pets or companions but wild animals in the broadest sense as well (cf. some texts in [19])—equally well. However, my aim in this text is not to combat my own (or anybody else's) anthropocentrism by way of widening my scope in *this* direction. Instead, I want to focus on how grief—both interpersonal and ecological—works in personally specific ways, conditioned by how one understands one's loss, and on how the mark that one's loss has left on one's understanding of one's life is also (if not centrally) the matter of what options for one's life one perceives (or fails to perceive): in particular, the eroded sense of the future. These two lines of analogy will be explored in the following two sections. The human (mostly) contexts I rely on as the

springboard simply provide me with a rich reservoir of valuable observational points; that cases of grief after non-human animals would work similarly seems obvious to me, but arguing for or against this similarity is neither my ambition nor—I believe—needed for the point I want to make.

Similarly, much valuable work has been conducted on responding to grief by way of meaning reconstruction [20] or relearning the world [21], which is vitally relevant in view of grief being the matter of what options are open (intelligible) to the grieving person. In fact, ecological grief may be energising and mobilising the griever towards active work in conservation efforts, climate activism or elsewhere. I am leaving these mostly aside not in order to deny their relevance but because my primary interest is with cases in which this does not (quite) take place. For it might be deceiving to suggest that there is an organic, essential link between experiencing ecological grief and a drive to such active engagement; just as there is no such *essential* link between grief and a loss of a drive to active engagement. My comparatively greater interest in cases of this latter kind should not indicate an argument (if implicit) in favour of such an essential link. Grief, as I mentioned at the beginning of this section, referring to Michael Cholbi's observations, takes many different forms.

Please, let the following remarks be read in the indicated light.

### 3. Ecological Grief as Personal

The authors introduced in Section 2.1 would agree that the particular shape that grief takes differs strongly from person to person. Although they identify some "structural features" of grief that occur more or less generally, these have quite varied expressions. For instance, as Cholbi notes, Kübler-Ross's famous five-stage model (denial–anger–bargaining–depression–acceptance) may or may not occur: people often skip some of those steps, reshuffle them, become frozen before arriving at the "end", and so forth. The more general structural features are such that can produce very different pictures: being personally invested in somebody and losing not only the person but also the possibilities of one's own identity and life that the person has represented can, of course, manifest in very different ways.

There seems to be a slight difference with ecological grief: most philosophical accounts of grief (as briefly outlined above) work with the pattern within which the focus is on another *person* and this deceased person is *central*, or at least very important, to the bereaved person's identity, self-conception, and life, and the bereaved person is well aware of it (if not in as many words, then at least implicitly), which is why the impact on the whole world of the bereaved person is so shattering. On the other hand, places, landscapes, biological species, and ecosystems mean different things to different people. It is possible to be aware of the fact that the woods in the vicinity of one's hometown are suffering from pollution and disordered climate conditions and that their biodiversity is irreparably damaged as a result and yet not be significantly "moved" emotionally by it. A lot depends on whether one lives at the forefront of the climate crisis as well as on other factors; the "natural response", as Cunsolo and Ellis characterise ecological grief, does not inevitably mean "necessary". To say that one is well aware that one's spouse or close friend has died yet one is not notably moved emotionally by it sounds strange at least (including, of course, emotional responses such as numbness or emptiness; as has been said, the particular phenomenology of grief experiences is varied and includes also this kind of "inhibited grief" [9]).

This indicates that in ecological grief, a greater role is played by the uniquely personal investment of the person in the object of grief, and this investment may require a special relationship, time investment, or active thought spared on the subject. The paradigmatic cases of ecological grief in people who live on the frontline of the climate crisis—as reported by [1,22]—show that environmental degradation is a direct blow to one's identity, as identity is closely tied to practices (such as fishing) that are simply not possible any more [22] (p. 475ff). Nowadays, for people living in the urban landscapes of developed industrialised societies, this link is less than obvious: the practices of their living, forming their identity,

may or may not relate directly to the environment. They are mostly not "people of the sea ice" (or something similar), and even if they conceive of themselves as "people of paved streets" (or something similar), the urban landscape is not degraded or disrupted by the effects of climate change in quite the same way as the sea ice. That cities are becoming "heat islands", disproportionately affecting those living in poorer and underprivileged urban neighbourhoods, is likely to induce (politicised) anger and a heightened sense of climate injustice (as has been shown, e.g., by Hickman and her colleagues [23]) rather than an overwhelming kind of grief similar to that which we encounter among Cunsolo's Inuit respondents.

This is not to say that a well-off person living in a city and still relatively sheltered from the most disruptive effects of climate change cannot experience ecological grief. It can be provoked even by rather parochial "epiphanies": for example, the perceived gradual disappearance of snow from Central European cities (in non-mountainous areas) in winter. When I was a boy, it was perfectly normal to use a sledge for whole weeks in winter. Now that people of my generation have kids, they duly buy sledges for them, but this once indispensable item of equipment now mostly stays in the closet: there is just not enough snow in winter. And people usually need to take organized trips to the mountains, but that is mostly to ski; sledging as a low-key pastime that can be undertaken on the spur of the moment and just for an hour or two has more or less vanished from my socio-geographical context. I would like my daughter to experience what I experienced as a child, but it does not seem possible any more. This is a loss of certain life possibilities not only for her but, vicariously, for me as her father, too (this experience has been called "snow grief" [24]). At the same time, the lack of snow in the city is not simply the loss of something I once had in my life, while she does not. There are many such things (unlike me, my daughter also does not have a grandmother with a large country cottage with chicken and rabbits); this one, however, functions as a focus (cf. [25]) concentrating my sight and allowing it to see something broader—the absence of city snow—in a particular light: as a systemic environmental loss over which to grieve (to see means to respond emotionally at the same time) [26,27].[2]

This idiosyncratic, parochial example shows that while ecological grief cannot rely on a reasonably guaranteed centrality of personal investment in the lives of everyone (compared to grief for one who is close to us), it allows, on the other hand, for great plasticity in what can awaken the emotion, how it can be awakened, and in what context. To put it more simply, there is probably very little that would more or less *have to* motivate ecological grief in everybody, *ohne weiteres*. Even the most straightforward cases, such as those reported by Cunsolo, are, in fact, cases of a complex interaction between environmental degradation and the particular *identity* of the people, connected and rooted to the place. But there are very many, often quite surprising, impulses that *can* motivate it—in the right circumstances. Notably, some of these impulses, in order to work, require some investment of reflection or consideration on the part of the one who experiences them or particular idiosyncratic contents of personal history. The relatively complicated trajectory featuring children's sledges—as outlined in the previous paragraph—could not happen automatically to just anyone in my position. I need to have spent some time, in the past, thinking about, reflecting on, and paying attention to the connections that formed the background to the way the above insight struck me. This is not the same as furnishing this striking insight by explicitly and deliberately thinking (about) the connection, thereby thinking it "into" existence.

Martha Nussbaum introduces a very personal example of her grief over the death of her mother to show that, much as the emotion involves tumultuous and almost crushing feelings, it would be impossible for us to have those feelings without the sense of the object of the emotion, of what it is about. In her case, it is an emotion that is "about [her] mother and directed at her and her life" [28] (p. 28). Grief would hardly even be there without the sense of what it *means* to lose a close, beloved person. We grieve genuinely even when we are not in direct contact with the person and even when we perhaps hear about her death only months later. Nussbaum's account shows that the experience of grief

does not differ significantly whether we grieve spontaneously and immediately or not. At least *some* cases of grief do not involve being directly hit with the facts of a situation but rather spontaneously responding emotionally to something one (implicitly) recognises as important. Feeling grief is connected to this recognition.

Ecological grief exemplifies well these complicated relationships: some people simply do not experience ecological grief even when they know just as well what is happening. This absence of grief amounts to their not responding to the loss as to a loss of something that is of value to them (this is, of course, not either–or: grief admits different degrees of strength, corresponding to the strength of our emotional attachment to its object, now lost). Nussbaum's account of emotions is notably cognitivist, but my emphasis, in construing ecological grief in analogous terms, would not be on the very knowing of what is happening but on understanding or appraisal, by way of a practical attitude, what the events mean, what kind of value, or what kind of threat to what kind of value, they represent (in relation to the role the perception of value threatened plays here, cf. my take on environmental grief as an "ailment of the soul" [29]). I am thus not trying to side with any particular theory of emotion; after all, some writers would plausibly argue that the ambition of a unified theory is more harmful than helpful, since each theory explains well the features of emotional phenomena that it emphasises and much less well other features emphasised by other theories [30] (chap. 2).

My partial emphasis on some aspects of ecological grief is motivated by the assumption that in order to do it justice, we need to focus on those respects in which it is relatively far from being an instinctive, automatic gut feeling and seems to rest and elaborate on the recognition of certain values (or of certain facts in a certain light). (In general, this is in line with the observations that grief as such has a complex arc and differs from more "episodic" emotions.) There are (at least) two senses in which this non-automatic character is at play. One of them—perhaps the more important, on the whole, though I am mostly leaving it aside in this text—concerns the complicated mechanisms through which what counts as *grievable* for us at all; traditionally, the domain of the non-human (animals, but not only them) has been excluded from the grievable [19] (p. 15f). (Re)establishing the neglected parts of our world may need applying particularly adjusted rhetoric [31] and, inevitably, amounts to a *political* agenda. Secondly, on the background of these deep-seated patterns of recognised grievability, our capacity still takes, on top of that, quite person-specific forms, in response to very particular, even idiosyncratic triggers. These highly personalised triggers work, however, thanks to there being the background of our cultivated sensitivity towards certain situations as loaded. This makes our emotional responses of ecological grief both personally varied and, yet, broadly mutually relatable, thanks to largely overlapping patterns of grievability.

Some grieve after witnessing a polluted river; some after witnessing starving stray dogs fighting for pieces of plastic to eat; some after realising that their children will have less snow and winter fun in their lives; some grieve directly for the loss of what the trigger situation is about; some come to realise, through their response, something more systemic or indirect and grieve for that. And yet, all this variety is apparently well compatible with the fact that ecological emotions are regularly and probably plausibly approached as such that can be judged as justified or less justified [32]. (And they mostly *are* justified.) Some appear to have good reasons, some have less good reasons. Even less complicated and more immediate emotions—such as the proverbial fear of a threatening bear—are liable to this evaluation. However, unlike fear, which, even if unjustified, remains fear (stronger versions of cognitivism might not concede even that, though; cf. Solomon on anger [33]), ecological emotions are vulnerable to depreciating reclassifications: if a critic succeeds in convincing the distressed person that their ecological grief is "confused" in view of the supposed facts that should justify it, it is far too easily relabelled as panic, hysteria, or some kind of neurosis, and thereby disenfranchised [1,9]. The always contestable issue of legitimacy seems embedded in the very heart of ecological emotions or at least some of them. Grief appears to be one such case, perhaps in a difference from hope or anxiety—there can be

such a thing as "deluded hope" or "neurotic anxiety", but can there be "deluded grief"? For sure, even cognitivist accounts would mostly satisfy themselves with what the one who feels the emotion *thinks* to be the case rather than requiring that it be the case in order that the particular emotion counts as such. However, ecological emotions clearly are more sensitive to this cognitive aspect both in order to *pass* as *these* emotions and because of their looped character. The explicit awareness of the recognition of the climate crisis as *correct* fuels both these emotions in those who experience them and the denialist stance (real or perceived) of those who do not share them. And those who do not share these emotions (who do not admit the underlying assumption of facts recognised) can thus either threaten or exacerbate the emotions (in the case of anger) or at least constantly force ecological emotions to face this resistance, coexist with it, and cope with it.

We may perhaps say that because of the sensitive political nature of climate change, environmental protection, and so forth, and because of heated public debate, ecological emotions have emerged, have been constructed, and keep being cultivated such that their own self-conception places some emphasis on this cognitive aspect and such that at their heart stands the concern for recognition. They thus serve as a good example of the observation made by Rorty that emotions "do not form a natural class", perennially human and universal, and that their list instead grows, expands, and transforms with time and the historical developments of culture(s) [34] (p. 141). Ecological grief is a good example of an emotion that has emerged and established itself as such as a result of a complicated historical development [35,36].

My take on ecological grief can thus be summed up as follows: the motives we have for how we feel (or simply the "why" of why we feel how we feel) can be reconstructed as *reasons*—and by "reconstructed" I do not mean "fabricated"[3]—in the light of which how we feel plausibly appears as more or less justified, to an open range of observers, including both the first person and third parties [37]. This kind of evaluative assessment is increasingly applied to environmental emotions [32]. The weight of these reasons can vary quite fluidly from person to person. My ecological grief is motivated by (and reflects) how I understand or consider my life in relation to the lives of other people or non-human others. Roughly the same can, of course, be said about the much more striking, thoroughgoing and direct forms of ecological grief present in Indigenous communities on the frontlines of the climate crisis. But in both cases, it means something quite different: not only does it look very different but the mechanism of the motivation can hook onto very different kinds of life contents: on how one earns one's living, on what one owes to one's community, on what one wishes for one's children, on what one finds beautiful (as one respondent in Ashlee Cunsolo Willox's film [38] has stressed), or on what one cherishes about one's childhood memories (whether or not they are of direct practical importance for one's life now).

The "reasons" (broadly construed) that we have for grieving for what has been lost or damaged in ecosystems can be very idiosyncratic, such as my anecdote about childhood sledging. The way in which we recognise these reasons can be idiosyncratic as well, and our practices of grieving (acting upon these motives) can also be idiosyncratic. None of this is to say that these reasons (and, by extension, our ecological grief) are not legitimate or genuine, nor that the discernment between emotional responses justified and unjustified could not apply.

## 4. "Moving On"?

Earlier, I pointed towards the *variety* of grief even in interpersonal contexts: the "structural" description of grief as a personal (identity-related) investment in someone and a corresponding reaction to losing that person, along with everything she represented, for one's life and identity allows for grief to take various shapes. Some of these differences can show themselves in connection to what happens *after* grief: when the grieving person comes to "closure" or manages to "move on". By this, I mean what overall shape the person's life apparently takes, in the eyes of an impartial if sympathetic observer, much as it need not look like that from her own perspective. (There are still a lot of conflicted

feelings in a person whose life has, apparently, "normalised itself" again.) Grief develops in ways that can be "narrated" (*pace* Goldie) intelligibly to a listener, and "moving on" certainly belongs among the more conventional (not necessarily in a bad sense) structural features of such narratives. I will try to show that even the presence of this feature can be problematised, though, again, in intelligible ways.

Closure by way of "moving on" comes with different degrees of "finality" (and is itself a debated topic in grief research [39]). Those who have lost an elderly parent experience a blow, but *often* (for sure, not always), this loss nevertheless eventually comes to *feel* as a natural event in one's life and as something one needs to (and usually does) come to terms with. Losing one's spouse or life partner has a different phenomenology. It is a blow, too, likely more severe than losing one's parent. While losing a parent may force one to see that one is now irrevocably an adult (from a perspective that one may not have fully appreciated until that moment), losing a spouse may reveal something about one's own finality of life rather more dramatically. The experience of this kind of loss also, perhaps depending, to an extent, on how old one perceives oneself to be, can open the intriguing dilemma of whether or not to "remarry". Losing a child comes much closer to the experience of something unnatural and unspeakable; here, both the reality and the very idea of "moving on" may strike one as either unintelligible or desecrating (as Denise Riley's notes indicate). Losing a close friend is a blow that does not inherently resemble closely any of these, and "moving on" does not rely on replacing the lost friend with another who would assume the "same" capacity—although relationships with other close friends can help. (All these distinctions are schematic, and their fit with real particular cases will naturally be limited.)

In most cases, though, where any kind of "moving on" can be observed, it usually takes the shape of the subsiding of the paralysing emotional experiences that blocked one's capacity to function in everyday contexts, the acceptance (often ritual) of saying "goodbye" to the deceased other, or of resuming one's investment in one's life independently of the deceased (sometimes, particularly in cases of remarriage and analogous, thanks to forging a comparably strong personal relationship with another special person). At any rate, whatever shape this moving on takes, it is quite intelligibly taken to mark the end, or at least the resolution, of grief, which is in line with Cholbi's observations. And, conversely, if there is one distinct sense in which the person can be identified as unable to "move on", it may indicate that she is still somewhat "stuck with" her grief.[4] Ecological grief may be exactly in this sense "complicated" or "prolonged" [9], resembling "continuing bonds" in bereavement [40], though compared to the variety and richness of this phenomenon in interpersonal contexts, the experience of ecological loss may more strongly tend towards *resisting* a positive perception of the continuing bond (the unequal applicability of the—hardly ever neutral—label "living in the past").

What kind of moving on is there in cases of ecological grief? Much as the idealised picture of "moving on" as before does not hold water and bereaved people typically need to rebuild their lives in a different shape than before, moving on still involves resuming one's life and re-engaging with many (if not all) of one's previous (everyday) activities and engagements. There are forms and aspects of (some cases of) ecological grief with which this idea sits uneasily. As Read points out, "while a healthy reaction to grief over a lost loved one is to grieve deeply and then gradually to recover, there is no 'recovery' from ecological grief" [2] (p. 103).

For one thing, when it comes to ecological grief, there is certainly no possibility of remarriage: ultimately, we have only one planet and we will never have another, and perhaps the same would apply to "the beautiful crystal clear blue lake of our childhood years". But, of course, recovery from grief can take various shapes. It need not just be the rebuilding of one's life in a largely similar shape but simply finding a way of re-engaging with one's life and overcoming the initial paralysis (see here [41] for the discussion of the complexities of recovery in ecological grief).

However, there seems to be little "space" into which one can "project oneself" to overcome the paralysis when we experience ecological grief of a "global" scale. Here, it

may amount—in some, not untypical, versions—to the realisation (whether realistic or just experienced) that the whole space into which one otherwise could project one's life has been eroded, both pragmatically and existentially (to the extent that one understands one's life as rooted in a home that has a certain stability, continuity, and potential). This may complicate the idea of "moving on" in cases where ecological grief responds to a systemic erosion of the *whole* space of resources from which we usually draw in cases of "moving on". By the "whole space", I do not mean all the material resources of life (though it need not be unrelated)—the whole world's ecosystem—but rather whatever kinds of resources one relies on for *seeing* how to go on with one's life.

Admittedly, many cases of ecological grief respond to a local and localised loss (or to the loss of only some part of important resources perceived as relevant to the good life) and instead energise the griever to a focused activity to try to make a difference (through conservation works, community work, political engagement, climate activism, …). I do not want to overlook the importance of these mobilising cases of ecological grief; however, my interest lies elsewhere, for it seems to me that the demobilising cases of ecological grief should not be dismissed as in any way secondary, for, I believe, they exhibit just as striking lines of analogy with much that we understand as central in not untypical cases of personal grief. The distinction between mobilising and demobilising grief I allude to here is not unconnected also to the varieties of *hope*, which are alluded to as the natural counterpart of distressed ecological emotions. While some authors discuss hope as a realistic assessment of options of action, available and perceptible even now [42], others stress that we may need to prepare ourselves for being able to work with our hope towards a future (as yet) "unimaginable" for us [43], cf. [2], in a sense similar to Jonathan Lear's notion of radical hope [44]. In the latter sense, ecological grief—whether perceived as global or local—may not be mobilising but rather represents something irreparably broken in one's life (one's life take on one's life) in relation to which any future-directed endeavour feels as a kind of acting against the stream. Rupert Read recalls Kierkegaard's notion of "hope against hope" in this context.

It is in light of these considerations that I think it is useful to ask how to understand Read's observation that there really is no recovery from ecological grief. There are indeed several options for how to understand this. Perhaps the most naturally healthiest is to learn to *live with* one's grief rather than being stuck in attempts to "move on" that are doomed to fail. Joanna Macy's project *The Work That Reconnects* can serve as an example of transforming grief into positive engagement. I am, however, perhaps more interested in options testifying that one *cannot quite move on* after the ecological loss, and yet it may fall short of a healing (and healed) practice. One obvious option here would be the image of someone struck by chronic depression, someone who lives their life permanently under a thick blanket of grief—grief of the kind that prevents them from everyday functioning (as happens quite typically in the early stages of bereavement). If ecological grief meant this, it would truly be alarming, primarily in the sense of a threat to public mental health (Scholarship on ecological emotions has occasionally explored the topic as a potential public health issue cf. [1,45–47]. More recently, though, there have been voices suggesting a need to go beyond the framing in terms of mental well-being [47]). In fact, ecological grief is alarming rather as a pointer towards massive, existentially threatening environmental degradation, and yet, even as such a lucid realisation, it simply does not need to take the shape of permanent paralysis, even when people do not really recover from it. How so?

The existing accounts of grief, such as those referenced in Section 2, commonly agree on an eroded sense of time accompanying grief, which is often combined even with a certain animosity towards (the very idea of) proceeding towards a future. Consider this fascinating remark by C.S. Lewis:

> And grief still feels like fear. Perhaps, more strictly, like suspense. Or like waiting; just hanging about waiting for something to happen. It gives life a permanently provisional feeling. *It doesn't seem worth starting anything*. I can't settle down. I yawn, I fidget, I smoke too much. Up till this I always had too little time. Now

there is nothing but time. Almost pure time, empty successiveness. [15] (p. 670; my emphasis)

In this case, we witness, to be more precise, a heightened awareness of time, but combined with a strange unwillingness or incapacity to *function within* time. One has a lot of time, but time seems to have lost the meaning of the "space" (?) in which new projects can be started and developed. Usually, this space lies in the future.

Riley's refusal to admit the prospect of "moving on" is even more specific:

To tell someone with a dead child, 'You should move on,' is doubly thoughtless, because there's no medium left through which to move anywhere. We were drifting through our former time like underwater creatures furnished with gills that they didn't notice they had, until they were fished up out of their element and their breathing apparatus failed.

If there is ever to be any movement again, that moving will not be 'on'. It will be 'with'. With the carried-again child. [17] (p. 41f)

As Riley's notes document, when the re-entry into social life eventually does happen, almost by itself, the grieving person may experience it as profoundly sad, as something she is not at all happy about (p. 80).

Cases such as these suggest that grief often brings about a certain unintelligibility of "moving on", which naturally may, but also may *not*, involve the experience of painful, paralysing sadness. In his ground-breaking research into "bereavement visits", which dates back to the 1960s, the Welsh physician Dewi Rees has been exploring the phenomenon of bereaved people seeing, meeting, or talking to their lost ones [48]—which he, just as others, strives to describe as a natural and healthy phenomenon helping the bereaved people [48–50]; cf. again the discussion of continuous bonds both in [9] and independent of ecological contexts [12] (chap. 5 and 6), [40]. Some of the verbatim statements Rees collected from his respondents show rather particular forms of the arrested drive to "move on": for instance, a widow of 27 (sic!) years stated, "I often have a chat with him. That's why I've never bothered with anyone else" [51] (p. 270).

If anything, this (admittedly, very brief) statement suggests contentment (if not straight-forward happiness) rather than paralysing sadness. On the other hand, it also shows that the death of a close person indeed takes away certain options for one's future, rendering them unintelligible. It may sound a stretch to call cases like this "instances of grief", and I am not sure whether I would want to claim this as a thesis, but it serves as a pointer to the fact that the eroded sense of the future that marks and commonly accompanies grief, if temporarily, takes various shapes, the common denominator of which is—rather than paralysing sadness—the incapacity to invest oneself in certain otherwise natural-seeming options for the future or to perceive them as intelligible (I have characterised elsewhere [52] this capacity of ecological emotions to prevent the very intelligibility of some life options as "grammatically disruptive", using the term "grammatical" in a loosely Wittgensteinian sense).

What shape could this take in the case of ecological grief? One relevant phenomenon might be the recent—and only recently covered—surge in environmental childlessness [52–54]. The most systematic account is that of Mathilde Krähenbühl, who has conducted extensive research with many respondents [54]. At the heart of this phenomenon is, as Krähenbühl puts it, "a palpable incapacity of people to project themselves into the future." This sounds very much like the observations offered by Riley or Lewis in their grief notebooks. Can ecological grief mean taking people's futures away from them and incapacitating their capacity to imagine a future family as a possibility? Krähenbühl indeed occasionally frames her observations of her respondents' statements in terms of "imagining a future" or the incapacity thereof. However, it seems to me that this parallel would be incomplete. People who refrain from having children for environmental reasons often have vivid ideas of what the future might bring. The future has not been removed as something real from their thinking; they are, in fact, preoccupied with the future to a heightened extent. They

anxiously consider not only the environmental pressure of the future population but also the predictable (low) quality of children's lives under climate breakdown. They are also conflicted about the future and invest much thinking into weighing the pros and cons of future parenthood. In this sense, their situation probably better fits that of anxiety (which is how Krähenbühl indeed conceptualises it) rather than grief (on eco-anxiety, see [55]). These anxious considerations are a natural response to the "complete loss of everyday practical familiarity" of the world as a habitable place, a "disintegration of everyday belonging" [18] (p. 52), as one has lost the sense of a place into which one can project oneself and one's future.

The key difference between the above and what I am exploring is that in anxiety, as described by accounts of climate anxiety, those who suffer from it are actively struggling for their future: they are not indifferent to it. On the contrary, they are invested in it and experience their incapacity to project themselves into it as a serious blow. On the other hand, when grief does not mobilise one to work towards the future (in relation to which the past both serves as a memento and is, in a sense, left behind), it can take rather take the shape of a certain unwillingness and lack of interest in the future. In a sense, a significant part of the future has disappeared here from one's "mental map", to borrow Bernard Williams' phrase. One may even *struggle against* having to have anything to do with the future. At the same time, while the future is the source of some uneasiness or alienation, it acts through its absence. What Lewis or Riley occasionally identifies and verbalises is the failure of the future to represent a salient real possibility. Grieving people who have this experience—especially if they are not professional wordsmiths—do not name it; they just live, day by day, in the absence of a meaningful future. It is a part, or an aspect, of how they perceive *all* things, not just the future or future concerns explicitly.

This may suggest something other than environmental childlessness as a good example of demobilising ecological grief. Let me remind readers, once again, of the fact that the eroded sense of the future need not take the shape of extremely painful, paralysing sadness but of many various things that others, perhaps not quite justly, may criticise as the grieving person's failure to "move on". Let me quote Riley one last time:

> You live under the sign of the provisional. Often with faint amusement over little debates: do you unpack this coral dress from storage as if, when the summer arrives in a few months, you'll still be alive to wear it? Yes—but purely because you enjoy the zing of its colour today. [17] (p. 42f)

This is an entry made "two years after". I will dwell on the mentioned *amusement* for a bit.

I cannot rely here on any published sources but rather on my own experience of a person who has been active for some years in green politics but who now feels somewhat resigned (or burnt out?). Being a bit older than most of Krähenbühl's respondents, I am not in the position of one who is still just considering parenthood; I *am* already a parent, which occasionally fills me with anxiety but equally, if not more often, with a certain amused confusion over how I can take myself seriously as a parent or as someone working in a profession that assumes there is some continuity and planning in time. I do think that environmental degradation has indeed taken away from us a lot of meaningful options for the future, yet instead of thinking anxiously about the uncertain future, I still live much the same way as before, only I find it difficult to create serious, elaborate plans for the future. I find it difficult to take seriously all my everyday actions in the capacity of a parent, a partner, someone working in an academic job—all roles that presuppose some continuity and interconnectedness in time. I am "planted" into these situations and I act upon them, but the idea of working hard to build a career and then solid foundations for retirement, in which I will be able to watch, with satisfaction, my grandchildren grow up, strikes me as somewhat misplaced, even confusing, much as the outward shape of my life and my "achievements" would suggest otherwise.

Mind you, my ecological grief is very mild. I do not live at the forefront of the climate crisis; I *can* afford this mildly alienated amusement. Although a significant part of myself

(one which I tend to hide from others) refuses or has lost the capacity to "move on", my life is still moving on by itself and me with it. But people stricken with ecological grief much more strongly than I may struggle to embrace the future as something real. This can even take quite drastic forms of not being able to go on with one's life [56]. In all the various cases of this scale, we can witness a disrupted, perhaps irreparably, perception of the future as something to be taken seriously, to build upon. This is a disruption of the kind that renders some things unintelligible.

This is not to deny that in many cases, ecological grief can mobilise us to meaningful endeavour. However, it would be shortsighted to picture ecological grief, in *all* its forms, as a necessarily helpful and healthy experience, especially when it responds to a loss that is in no way natural or that has been inevitable and when this loss has seriously damaging consequences for future life (human and non-human) on this planet, consequences about the avoidability of which we should probably not have too high hopes. Unsettling forms of ecological grief, more straightforwardly reflecting this perception, deserve to be explored with just as much attention as those that reconnect us with our environment.

### 5. To Conclude

Grief can take many different shapes. We are socialised to understand this emotion through its "core" cases: when we grieve for a close person who has died. Yet there are forms of grief that do not presuppose a close personal relationship, a person as the object of grief, or even death involved—see the studies assembled in [57], esp. [4] in relation to ecological grief—and therefore the parallel should be drawn with some caution. Ecological grief serves as one such example of grief, the nature of which is not personal and the object of which is undergoing serious damage, but it would perhaps be simplifying to call this damage "death" ("loss" is certainly more adequate). The loss may dawn on the person gradually or remain unnoticed or misunderstood for a long time. On the other hand, this shows that some aspects of ecological grief are better captured if we work with cognitivist-akin descriptions of the emotion (without having to subscribe to cognitivism as a general theory), as it both depends on and is facilitated by whether and how (in what terms) we understand the loss. Related to this are the aspects in which personal grief and ecological grief can be paralleled more safely: in tracing how both these losses represent significant blows or degradations to (existential) options of the grieving person's identity and life and their development. Instead of affective experiences more narrowly construed, such as the painful and paralysing feelings of sadness, I focus in particular on the eroded sense of the future as real, meaningful, or to-be-taken-seriously, which accompanies some cases of grief experience quite characteristically. This allows us to distinguish them from eco-anxiety (which is preoccupied with the future in a very different sense) but also to conceptualise, as ecological grief, various cases of failure (or refusal) to "move on", which otherwise may not fit the common images of grief, for instance in their relative absence of painful affective states. The phenomenology of (personal) grief, as offered by the rich and valuable first-person accounts of grief, provides some support for this reading. On the other hand, ecological grief is more closely tied to the sense of a place under threat—though place not simply in these of physical geography, but of one's home, tied to one's sense of identity—it also exhibits a (geographical) variety, dependent on how far from the forefront of the climate crisis one lives [58]. This kind of variety may not find any straightforward parallel in the varieties of personal grief.

**Funding:** This research was funded by project no. 22-15446S, "ECEGADMAT", of the Czech Science Foundation.

**Institutional Review Board Statement:** Not applicable.

**Informed Consent Statement:** Not applicable.

**Data Availability Statement:** No new data were created or analyzed in this study. Data sharing is not applicable to this article.



**Acknowledgments:** Laura Candiotto read a draft of this paper and provided me with a number of helpful suggestions. I am indebted to my reviewers, whose comments prompted me to consider further aspects of the topic and who also significantly expanded my knowledge of existent research pertinent to ecological grief.

**Conflicts of Interest:** The author declares no conflict of interest.

## Notes

[1]  Rupert Read mentions the Moore's Paradox-like form of grief [2] (p. 96ff).

[2]  This example also shows how fluid and porous the categories of particular ecological emotions are—here, I am coming close to what Lertzman characterises as "environmental melancholia" [26] and Bruce Albrecht as "solastalgia" [27].

[3]  This notion of reason that I use—what we (may *need* to) reconstruct as an (intelligible) answer to the question "Why?"—is not dissimilar to the famous way in which Anscombe analyses "intention", along with the importance of the description under which we understand (and perceive) what is going on [37].

[4]  The TV series *Monk*, about a genius but obsessive investigator also offers, at a secondary level, a detailed long-term study of grief. The titular character, who has lost his wife to an—unsolved—murderous attack, is seen, for many years, as keeping many of his wife's belongings, including drawers full of her clothes, and sleeping on one side of the bed only while the other remains "reserved" for her. All that stays with him, although he has managed throughout the years, more or less successfully, to reintegrate into his professional life and to rekindle and cultivate personal friendships.

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
