# Peer review of "Ecological Grief Observed from a Distance"

_philosophies, doi:10.3390/philosophies9020037_

Round 1

Reviewer 1 Report

Comments and Suggestions for Authors

This manuscript's authors presented a coherent concept of ecological grief, setting it against the background of personal grief. The structure of the manuscript, the precision of the language, and the caution of the assessments indicate the high scientific competencies of the authors. The authors' theses are interesting, original, and adequately justified.

I am pleased that philosophical studies that include emotions in the discourse on the environmental crisis are being developed. In this respect, grief is an important factor in expanding the contemporary debate on this topic. Similar to the emotion of shame, which also appears in philosophical reflection on environmental issues, see Klimková, A. (2023). Is Shame Morally Relevant to Ecological Ethics? Studia Ecologiae Et Bioethicae, 21(2), 5–13. https://doi.org/10.21697/seb.2023.15.

Author Response

Thank you for reviewing the manuscript and considering it worth publishing. In the end, I have not included a reference to the paper you mention, as it is relatively distant topic-wise, but I may use it elsewhere.

Reviewer 2 Report

Comments and Suggestions for Authors

Thank you very much for this paper. I really enjoyed the read.

I only have minor comments in the text: one typo and some questions on the theory. In the beginning you give a definition of grief by Cholbi but you don't seem to use this throughout the article. I think you could structurally improve the paper by using what you expose in the beginning.

I have just one concern: The special issue of the paper that you want to publish in is about moral emotions. Your paper does not explicitely speak about moral emotions, does not cite the literature and does not argue for the morality of ecological grief. So I am not sure, whether you chose the perfect place to publish, but I guess this is for the editors to decide.

Comments on the Quality of English Language

there is one typo, other than that no remarks on the language

Author Response

I have clarified the notion of grief that I am using (as such that intersects with various points stressed by the authors I refer to, but does not tally with their accounts on the whole): "a complex response to a recognised loss of a value important or even central to the grieving person’s life (which can subsequently assume the meaning of something left, as it were, in the past)", a response that "involves a transformation of what the whole world, the whole life, means to the grieving person, especially with respect to what existential options appear as open or relevant to her". I also stressed the heterogeneity of grief (my observations about ecological grief naturally pertain some of its forms but not necessarily others).

---

It is true that grief is not a moral emotion specifically (though the special issue is not about moral emotions as such, but about moral psychology of emotions - but your concern is valid here as well). I could have stressed the link to more specifically moral emotions in the ecological family (guilt, shame), but eventually I was following a different line of discussion where this did not play a role. (However, I was encouraged by one of the special issue's editors to propose an abstract and it was approved for the issue - this is a small, if "external" justification.)

---

I have gone through the particular comments inserted in the pdf and tried to accommodate as many of them as possible. They were helpful for me.

Reviewer 3 Report

Comments and Suggestions for Authors

Ecological grief is a highly important topic and it is a subject of growing research interest. However, there are still not many philosophical in-depth discussions of it (although more than what is cited here, about which more below). This manuscript provides a very interesting application of grief philosophy into ecological grief. It needs further links with existing research, but otherwise it is close to being able to be accepted for publication. I would very much like to see it published after revisions.

The following article collection would be important to consult and cite:

Cunsolo, A. and Landman, K. (eds) (2017) Mourning Nature: Hope at the Heart of Ecological Loss & Grief. Montreal & Kingston: McGill-Queen’s University Press.

Pihkala, P. (2024) ‘Ecological Sorrow: Types of Grief and Loss in Ecological Grief’, Sustainability, 16(2), p. 849. https://doi.org/10.3390/su16020849.

Barnett, J.T. (2022) Mourning in the Anthropocene: Ecological Grief and Earthly Coexistence. East Lansing: Michigan State University Press.

Via these sources, please consider how losing non-human animals can be, and has also been found in research to often be, a significant loss. Some of the formulations are quite anthropocentric.

A few broad points before more exact comments. You are focusing on your applications on rather broad-scale ecological grief. Some formulations work better for that, and not as well for more particular and local ecological losses and griefs. I have made suggestions below to make this better. It would be good, however, to tell the reader your focus a bit earlier: something of the content on lines 616-9 should be mentioned in the Introduction.

This is a journal of philosophy, but still I think that some more concepts from psychology should be mentioned. Some examples are given below. Overall, I think that the depth of emotional attachments is significant for the content (cf. grieving a bit of a distant grandfather, contrasted with grieving a beloved grandmother).

Comments with line numbers

Abstract

6 add “MANY philosophical accounts…”

7 ”even identity”; I do not think that it is very extraordinary that grief has links with identity. Please consider rephrasing and connecting this with the concept of intangible loss.

The parentheses can easily be removed, and should be.

9-10 ”a loss severe enough to intelligibly threaten her life and the world as meaningful and sustainable wholes.”

-> A broader point: please consider that ecological grief may arise even when the ecological loss is not threatening one’s life as meaningful; it may “simply” be a very sad loss.

16 “grammatical disruption” is not understandable here without reading the paper. Since it’s not a very major point in the article, please consider removing it from here; otherwise, you need to explain.

Keywords: “anticipatory grief” would be a good addition

21-27 please consider marking the original sources of the quotes directly after them.

53-64 Please see and cite the historical review in Pihkala 2024 and make a few adjustments here. I think that the “ecological grief” experienced by indigenous peoples should be mentioned here, even while it was not called by that term for a long time.

In relation to Leopold, see Barnett 2022.

69 “make” -> may

Section 2.1. is well written and the author nicely highlights points that they are going to focus on later.

156-8 This summarizing observation should, at some point in the article, be at least briefly linked with what esp. Pihkala 2024 discusses in relation to Neimeyer’s work on meaning reconstruction and Attig’s work on re-learning the world.

174 “CAN make”, not “makes”! (Cf. your own lines 89-90)

181-4 The research on guilt in bereavement often discusses these kind of dynamics.

200 Please clarify what you mean by “fringe forms”. Who makes that judgement? Cf. the research on pet and companion animal bereavement, and posthumanistic ecological grief scholarship.

203-4 The Moore Paradox should be either explained or then left out.

211-2 This reminds me of grief researcher Worden’s observations about the need to adjust to a new kind of environment.

214 Please mention earlier in this paragraph the cause of Riley’s grief which she writes about.

226-9 The point is nicely put here.

234-6 Perhaps the names of the five stages in the DABDA model could be briefly mentioned. Many readers will be familiar with them, but not necessarily all.

245-53 This would be a good place to discuss the topic of emotional attachments.

249-51 This is sometimes called “inhibited grief” in grief research, and there’s a wide discussion about it. See e.g. Pihkala 2024.

267-9 Here would be a good place to add a source about climate injustice and/or climate anger.

273-288 this has sometimes been called “snow grief” or “winter grief”. See e.g.

Zimmer, Katarina (2021). The snowy countries losing their identity. (2021). BBC. https://www.bbc.com/future/article/20210215-winter-grief-how-warm-winters-threaten-snowy-cultures.

293-4 I understand what the author is after here, but I disagree: there are cases where the lost ecological feature is so significant for the community and individuals that it practically has to cause grief. See e.g.

Amoak, Daniel, Benjamin Kwao, Temitope Oluwaseyi Ishola & Kamaldeen Mohammed (2023). Climate change induced ecological grief among smallholder farmers in semi-arid Ghana. SN Social Sciences. 3(8), 131. DOI: 10.1007/s43545-023-00721-8.

One way to correct this is to add a qualification here: “ecological grief IN THE GLOBAL SENSE” or similar.

315-6 Here the “psychology of loss” framework would offer sources.

322-4 Even if certain theories are not named, I think that the concept of appraisal should be mentioned here.

338-40 The sentence seems to miss some words. Please clarify.

345-6 Here, it would be good to name disenfranchised grief. See Cunsolo & Ellis 2018; Pihkala 2024.

356 Why (“denialist”), repetition of the same word?

357-9 I would split the sentence to make it clearer.

360-9 I miss some links to earlier research here. See e.g.

Neckel, Sighard & Martina Hasenfratz (2021). Climate emotions and emotional climates: The emotional map of ecological crises and the blind spots on our sociological landscapes: Social Science Information. SAGE PublicationsSage UK: London, England, 60(2), 253–271. Sage UK: London, England. DOI: 10.1177/0539018421996264.

Pihkala, Panu (2022). Toward a Taxonomy of Climate Emotions. Frontiers in Climate. 3. https://www.frontiersin.org/article/10.3389/fclim.2021.738154. DOI: 10.3389/fclim.2021.738154.

365 This reference to theories of basic or universal emotions would benefit from clarification; it’s not supported by all emotion scholars. A simple add-on will do.

374 Please mention a source as an example.

376 “other people OR NON-HUMAN OTHERS” would be much better here. Cf. the studies by Cunsolo and colleagues on relationality between Inuit and caribou, and ecological grief.

393 add “common” grief, or similar, to make clear that you’re not talking about ecological grief here

405-> please add at least one source about closure in grief research. E.g.

Boss, Pauline & Donna Carnes (2012). The Myth of Closure. Family Process. 51(4), 456–469. DOI: 10.1111/famp.12005.

408-12 Slightly problematic example, since people who don’t have a spouse may be driven to engage with their finality of life via the death of their parent(s).

429 Please connect this to complicated grief and/or prolonged grief with naming that and using a reference.

429-435 Instead of an anecdotal example from a TV series, I would have preferred to see here a research-based discussion on memorizing and continuing bonds. Grief researchers have discussed these dynamics a lot.

440-8 The possible dynamics of “acceptance” or “closure” have been discussed in ecological grief research. Please see e.g.

Pihkala, Panu. 2022. “The Process of Eco-Anxiety and Ecological Grief: A Narrative Review and a New Proposal.” Sustainability 14 (24): article number 16628. https://doi.org/10.3390/su142416628.

446-8 In line with the author, I do not agree that there’s no “recovery” (as Read argues); there may not be total recovery, but there may be a fulfillment of tasks of grief so that life energies are rejunevated. See Randall’s application of Worden:

Randall, Rosemary. 2009. “Loss and Climate Change: The Cost of Parallel Narratives.” Ecopsychology 1 (3): 118–29.

451-2 Again, this argument seems to work better for global ecological grief than for local forms of it. There may be surprising environmental damages which produce shock and grief.

457 Instead of “realization”, I would argue that this is “belief”; not everything has been yet lost, or in the near future. But naturally the author is free to argue as they like!

469 Instead of “stressed this c.”, correct the argument to something like “explored the topic as a potential public health issue”. Otherwise, readers might get the impression that Cunsolo and others include chronic depression as a standard part of eco-grief.

506-12 Here you must mention continuing bonds. See Pihkala 2024 for discussion of that and eco-grief.

602 instead of “trained”, consider “socialized” or “usually driven”. People are often not trained at all in relation to grief.

604-5 For “non-death loss” and ecological grief, see also Pihkala 2024; and Kevorkian’s article in the volume referenced.

625-8 See Benham, Claudia, and Doortje Hoerst. 2024. “What Role Do Social-Ecological Factors Play in Ecological Grief?: Insights from a Global Scoping Review.” Journal of Environmental Psychology 93 (February): 102184. https://doi.org/10.1016/j.jenvp.2023.102184.

Author Response

Overall: thank you for the very thorough reading and for so many helpful suggestions. My replies to the particular points:

The following article collection would be important to consult and cite:

Cunsolo, A. and Landman, K. (eds) (2017) Mourning Nature: Hope at the Heart of Ecological Loss & Grief. Montreal & Kingston: McGill-Queen’s University Press.

Pihkala, P. (2024) ‘Ecological Sorrow: Types of Grief and Loss in Ecological Grief’, Sustainability, 16(2), p. 849. https://doi.org/10.3390/su16020849.

Barnett, J.T. (2022) Mourning in the Anthropocene: Ecological Grief and Earthly Coexistence. East Lansing: Michigan State University Press.

*** Done

Via these sources, please consider how losing non-human animals can be, and has also been found in research to often be, a significant loss. Some of the formulations are quite anthropocentric.

*** I agree. I amended the formulations in question, and I explained my relying on cases of interpersonal (human) grief not as motivated by their centrality, but simply because these accounts serve me rich material for comparison. (It is true, though, that most comprehensive accounts of grief in contemporary philosophy (as those referenced by me) simply presuppose human context as a given.)

A few broad points before more exact comments. You are focusing on your applications on rather broad-scale ecological grief. Some formulations work better for that, and not as well for more particular and local ecological losses and griefs. I have made suggestions below to make this better. It would be good, however, to tell the reader your focus a bit earlier: something of the content on lines 616-9 should be mentioned in the Introduction.

*** Done.

This is a journal of philosophy, but still I think that some more concepts from psychology should be mentioned. Some examples are given below. Overall, I think that the depth of emotional attachments is significant for the content (cf. grieving a bit of a distant grandfather, contrasted with grieving a beloved grandmother).

Comments with line numbers

Abstract

6 add “MANY philosophical accounts…”

*** Done.

7 ”even identity”; I do not think that it is very extraordinary that grief has links with identity. Please consider rephrasing and connecting this with the concept of intangible loss.

*** I have rephrased the passage.

The parentheses can easily be removed, and should be.

*** Done.

9-10 ”a loss severe enough to intelligibly threaten her life and the world as meaningful and sustainable wholes.”

-> A broader point: please consider that ecological grief may arise even when the ecological loss is not threatening one’s life as meaningful; it may “simply” be a very sad loss.

*** I have made this formulation in the abstract less strong; later, in the paper, I try to differentiate among cases of grief more clearly.

16 “grammatical disruption” is not understandable here without reading the paper. Since it’s not a very major point in the article, please consider removing it from here; otherwise, you need to explain.

*** removed here; explained later in the main text.

Keywords: “anticipatory grief” would be a good addition

*** Added.

21-27 please consider marking the original sources of the quotes directly after them.

*** Done.

53-64 Please see and cite the historical review in Pihkala 2024 and make a few adjustments here. I think that the “ecological grief” experienced by indigenous peoples should be mentioned here, even while it was not called by that term for a long time.

*** Done. I have rephrased the paragraph somewhat in light of the added references.

In relation to Leopold, see Barnett 2022.

69 “make” -> may

*** Corrected.

Section 2.1. is well written and the author nicely highlights points that they are going to focus on later.

156-8 This summarizing observation should, at some point in the article, be at least briefly linked with what esp. Pihkala 2024 discusses in relation to Neimeyer’s work on meaning reconstruction and Attig’s work on re-learning the world.

*** I have added this reference.

174 “CAN make”, not “makes”! (Cf. your own lines 89-90)

*** Corrected.

181-4 The research on guilt in bereavement often discusses these kind of dynamics.

200 Please clarify what you mean by “fringe forms”. Who makes that judgement? Cf. the research on pet and companion animal bereavement, and posthumanistic ecological grief scholarship.

*** I removed the phrase (admittedly, an unfortunate one).

203-4 The Moore Paradox should be either explained or then left out.

*** Moved to endnote. (Where I, however, believe it does not need a further explanation, especially for a philosophical audience.)

211-2 This reminds me of grief researcher Worden’s observations about the need to adjust to a new kind of environment.

214 Please mention earlier in this paragraph the cause of Riley’s grief which she writes about.

*** Added.

226-9 The point is nicely put here.

234-6 Perhaps the names of the five stages in the DABDA model could be briefly mentioned. Many readers will be familiar with them, but not necessarily all.

*** Added.

245-53 This would be a good place to discuss the topic of emotional attachments.

*** The paragraph slightly rephrased to take into account the heterogeneity of attachments.

249-51 This is sometimes called “inhibited grief” in grief research, and there’s a wide discussion about it. See e.g. Pihkala 2024.

*** Reference added.

267-9 Here would be a good place to add a source about climate injustice and/or climate anger.

*** Added.

273-288 this has sometimes been called “snow grief” or “winter grief”. See e.g.

Zimmer, Katarina (2021). The snowy countries losing their identity. (2021). BBC. https://www.bbc.com/future/article/20210215-winter-grief-how-warm-winters-threaten-snowy-cultures.

*** Reference added (thanks for drawing my attention to this notion).

293-4 I understand what the author is after here, but I disagree: there are cases where the lost ecological feature is so significant for the community and individuals that it practically has to cause grief. See e.g.

Amoak, Daniel, Benjamin Kwao, Temitope Oluwaseyi Ishola & Kamaldeen Mohammed (2023). Climate change induced ecological grief among smallholder farmers in semi-arid Ghana. SN Social Sciences. 3(8), 131. DOI: 10.1007/s43545-023-00721-8.

One way to correct this is to add a qualification here: “ecological grief IN THE GLOBAL SENSE” or similar.

*** I rephrased the place accordingly.

315-6 Here the “psychology of loss” framework would offer sources.

322-4 Even if certain theories are not named, I think that the concept of appraisal should be mentioned here.

*** Done.

338-40 The sentence seems to miss some words. Please clarify.

*** Clarified.

345-6 Here, it would be good to name disenfranchised grief. See Cunsolo & Ellis 2018; Pihkala 2024.

*** Reference added.

356 Why (“denialist”), repetition of the same word?

*** Rephrased.

357-9 I would split the sentence to make it clearer.

*** Done.

360-9 I miss some links to earlier research here. See e.g.

Neckel, Sighard & Martina Hasenfratz (2021). Climate emotions and emotional climates: The emotional map of ecological crises and the blind spots on our sociological landscapes: Social Science Information. SAGE PublicationsSage UK: London, England, 60(2), 253–271. Sage UK: London, England. DOI: 10.1177/0539018421996264.

Pihkala, Panu (2022). Toward a Taxonomy of Climate Emotions. Frontiers in Climate. 3. https://www.frontiersin.org/article/10.3389/fclim.2021.738154. DOI: 10.3389/fclim.2021.738154.

*** Added.

365 This reference to theories of basic or universal emotions would benefit from clarification; it’s not supported by all emotion scholars. A simple add-on will do.

*** Rephrased.

374 Please mention a source as an example.

*** Done.

376 “other people OR NON-HUMAN OTHERS” would be much better here. Cf. the studies by Cunsolo and colleagues on relationality between Inuit and caribou, and ecological grief.

*** Rephrased.

393 add “common” grief, or similar, to make clear that you’re not talking about ecological grief here

*** Rephrased.

405-> please add at least one source about closure in grief research. E.g.

Boss, Pauline & Donna Carnes (2012). The Myth of Closure. Family Process. 51(4), 456–469. DOI: 10.1111/famp.12005.

*** Added.

408-12 Slightly problematic example, since people who don’t have a spouse may be driven to engage with their finality of life via the death of their parent(s).

***I slightly rephrased the whole paragraph added a caveat about the schematic (but, I think, as such still illuminating) and therefore limited nature of the introduced distinctions. 

429 Please connect this to complicated grief and/or prolonged grief with naming that and using a reference.

*** Done.

429-435 Instead of an anecdotal example from a TV series, I would have preferred to see here a research-based discussion on memorizing and continuing bonds. Grief researchers have discussed these dynamics a lot.

*** Done; the TV series example moved to an endnote.

440-8 The possible dynamics of “acceptance” or “closure” have been discussed in ecological grief research. Please see e.g.

Pihkala, Panu. 2022. “The Process of Eco-Anxiety and Ecological Grief: A Narrative Review and a New Proposal.” Sustainability 14 (24): article number 16628. https://doi.org/10.3390/su142416628.

*** Reference added.

446-8 In line with the author, I do not agree that there’s no “recovery” (as Read argues); there may not be total recovery, but there may be a fulfillment of tasks of grief so that life energies are rejunevated. See Randall’s application of Worden:

Randall, Rosemary. 2009. “Loss and Climate Change: The Cost of Parallel Narratives.” Ecopsychology 1 (3): 118–29.

***I made the claim less strong and clarified in what sense or in relation to what aspect of cases of ecological grief Read's point would hold. 

451-2 Again, this argument seems to work better for global ecological grief than for local forms of it. There may be surprising environmental damages which produce shock and grief.

*** Rephrased.

457 Instead of “realization”, I would argue that this is “belief”; not everything has been yet lost, or in the near future. But naturally the author is free to argue as they like!

*** Rephrased.

469 Instead of “stressed this c.”, correct the argument to something like “explored the topic as a potential public health issue”. Otherwise, readers might get the impression that Cunsolo and others include chronic depression as a standard part of eco-grief.

*** Rephrased.

506-12 Here you must mention continuing bonds. See Pihkala 2024 for discussion of that and eco-grief.

*** Further references added.

602 instead of “trained”, consider “socialized” or “usually driven”. People are often not trained at all in relation to grief.

*** Rephrased.

604-5 For “non-death loss” and ecological grief, see also Pihkala 2024; and Kevorkian’s article in the volume referenced.

*** Reference added.

625-8 See Benham, Claudia, and Doortje Hoerst. 2024. “What Role Do Social-Ecological Factors Play in Ecological Grief?: Insights from a Global Scoping Review.” Journal of Environmental Psychology 93 (February): 102184. https://doi.org/10.1016/j.jenvp.2023.102184.

*** Reference added.

Reviewer 4 Report

Comments and Suggestions for Authors

Thank you for the opportunity to read and comment on this manuscript. In this paper, the author discusses several prominent accounts of grief, develops the idea that ecological grief is personal (and, thus, highly heterogenous), and considers how ecological grief may manifest in ways that do not conform to commonsense expectations for what grief/mourning look like (e.g., foregoing parenthood). As I understand it, the author’s aim (or one aim, anyway) is to understand both how ecological grief is and is not like other modes of grief—in particular, the sort of grief one experiences in the wake of the loss of another human with whom one was, in some way, ‘close.’

Currently, the author presents ecological grief as a personal, subjective response to ecological losses and transformations of various kinds. This is certainly in line with some research on ecological grief (e.g., Cunsolo), but it does not account for the ways in which attention to, understandings of, or responses to such losses and transformations are culturally or rhetorically conditioned. As a way of enriching the view presented here, I encourage the author to deal more fully with the idea of “grievability” as initially developed by Judith Butler and subsequently applied in the context of ecological grief (see Barnett, Mourning in the Anthropocene, as well as Cunsolo and Landman’s introduction to Mourning Nature). The author is already gesturing in this direction, though perhaps without realizing it, when they write, on page 7 of the manuscript, that “we need to focus on those respects in which [ecological grief] is far from being an instinctive, automatic gut feeling and seems to rest and elaborate on the recognition of certain values.” The question is, under what cultural and rhetorical conditions does any ecological element or process assume value such that its loss would be registered as a loss?

In the section entitled “Moving On?” the author claims that “ecological grief responds to a systemic erosion—although not a complete obliteration—of the whole space of resources from which we usually draw in cases of ‘moving on’” (page 9). Here the author reduces the complexity of ecological grief. It is not always the case that ecological grief is in response to systemic transformations, however. It can also be the case that one’s ecological grief is in relation to a very specific habitat or species, for example. Even as habitat loss and extinction are certainly connected to and even drive by broader transformations, one’s sense of grief may not include an acknowledgement of this broader context.

At another key moment in the section entitled “Moving On?” the author again reduces the complexity of grief by suggesting that “grief-stricken people … experience rather a certain unwillingness and lack of interest in the future” (page 11). Quite a bit of work on ecological grief emphasizes its relation to practices of care and support. Rather than rendering people “unwilling” to engage in future-oriented work, ecological grief often catalyzes a sense of care for that which has been or (in the case of anticipatory ecological grief) that which might yet be lost (here again, see Barnett, Cunsolo and Landman, and others). So, in addition to the sorts of examples that the author addresses in this final section, I would also suggest that ecological grief may manifest, at least in part, in active engagement with, for example, conservation work.

Comments on the Quality of English Language

N/A

Author Response

Currently, the author presents ecological grief as a personal, subjective response to ecological losses and transformations of various kinds. This is certainly in line with some research on ecological grief (e.g., Cunsolo), but it does not account for the ways in which attention to, understandings of, or responses to such losses and transformations are culturally or rhetorically conditioned. As a way of enriching the view presented here, I encourage the author to deal more fully with the idea of “grievability” as initially developed by Judith Butler and subsequently applied in the context of ecological grief (see Barnett, Mourning in the Anthropocene, as well as Cunsolo and Landman’s introduction to Mourning Nature). The author is already gesturing in this direction, though perhaps without realizing it, when they write, on page 7 of the manuscript, that “we need to focus on those respects in which [ecological grief] is far from being an instinctive, automatic gut feeling and seems to rest and elaborate on the recognition of certain values.” The question is, under what cultural and rhetorical conditions does any ecological element or process assume value such that its loss would be registered as a loss?

*** I have incorporated this point into the text: what makes grieving possible for us rests on two layers of assumption - 1) the (socially, culturally, politically) established background of what usually counts for grievable, and 2) the personal links of mattering, of the valuable, resting upon (1). My text, however, is primarily interested in how our responses of (ecological) grief are personally based and intelligible as such.

In the section entitled “Moving On?” the author claims that “ecological grief responds to a systemic erosion—although not a complete obliteration—of the whole space of resources from which we usually draw in cases of ‘moving on’” (page 9). Here the author reduces the complexity of ecological grief. It is not always the case that ecological grief is in response to systemic transformations, however. It can also be the case that one’s ecological grief is in relation to a very specific habitat or species, for example. Even as habitat loss and extinction are certainly connected to and even drive by broader transformations, one’s sense of grief may not include an acknowledgement of this broader context.

*** This is a comment that other reviewers made, too. I tried to do justice to this concern: to distinguish more carefully between responses of a "global" scale and those more localised, to which some differences in how positively and actively one can react to one's grief correspond, along with the options of how one hopes. 

At another key moment in the section entitled “Moving On?” the author again reduces the complexity of grief by suggesting that “grief-stricken people … experience rather a certain unwillingness and lack of interest in the future” (page 11). Quite a bit of work on ecological grief emphasizes its relation to practices of care and support. Rather than rendering people “unwilling” to engage in future-oriented work, ecological grief often catalyzes a sense of care for that which has been or (in the case of anticipatory ecological grief) that which might yet be lost (here again, see Barnett, Cunsolo and Landman, and others). So, in addition to the sorts of examples that the author addresses in this final section, I would also suggest that ecological grief may manifest, at least in part, in active engagement with, for example, conservation work.

*** Again, a point made by other reviewers as well and one which I  did not consider sufficiently in the submitted version of the article. I now mention explicitly the heterogeneity of reactions to grief and refrain from formulations that might suggest that a failure to "move on" is an inherent characteristic of each and every case of ecological grief. What I tried to stress: it is an organic possibility in some cases, just as it is an organic possibility in some cases of interpersonal grief (a failure to "move on", in one way or another, is in no sense a deficient, pathological, or less healthy phenomenon than the opposite). Accordingly, a failure to "move on" after bereavement, even when unaccompanied by strong feelings of sadness and similar, can be understood as a form of (prolonged) grief; and there are good reasons to understand analogously such failures to "move on" in cases of ecological grief as well.

Round 2

Reviewer 4 Report

Comments and Suggestions for Authors

Thank you for the opportunity to read the revised version of this manuscript. I appreciate the authors’ attempts to address the concerns that I raised in my first review. Although I still feel that the cultural, political, and rhetorical conditions that give rise to and structure relations of eco-grievability are undertheorized in relation to the “personal” experiences of grief/eco-grief that the author takes up here, I understand that they are pursuing a different line of inquiry. In future work, I encourage them to consider the ways in which the “personal” is perhaps not so easily separable from the cultural, political, and rhetorical. Without this element, I think we miss something important about why and how our “personal” experiences take particular, often patterned forms. Nevertheless, I think this essay does advance our understanding of ecological grief in small ways and encourage the editors to publish it in its current form.

Comments on the Quality of English Language

n/a